

# Comparative analysis of biomechanical characteristics between the new Tai Chi elastic band exercise for opening and closing movement and elastic band resistance training for the reverse fly movement

Mingyu Liu[1,*], Cuihan Li[1,*], Xiongfeng Li[1], Jianwei Zhang[1], Haojie Li[1], Yameng Li[2], Qiuyang Wei[3], Zaihao Chen[1], Jiahao Fu[4], Yanying Li[5], Meize Cui[1], Lujia Li[1], Peng Zhang[1], Yuerong Huang[1], Yuxin Ma[1], Jianan Xu[1], Shaojun Lyu[1] and Yunchao Ma[1]

[1] College of P.E. and Sports, Beijing Normal University, Beijing, China
[2] Department of Physical Education, Northwestern Polytechnical University, Xi'an, ShanXi, China
[3] Sports Department, Jiangsu University, Zhenjiang, Jiangsu, China
[4] College of Physical Education, Zhejiang Guangsha Vocational and Technical University of Construction, Jinhua, Zhejiang, China
[5] Sports Department of Leisure Health Teaching and Research Section, Hainan Medical University, Haikou, Hainan, China
* These authors contributed equally to this work.

Corresponding authors
Shaojun Lyu,
l13121860699@163.com
Yunchao Ma, myc@bnu.edu.cn

## ABSTRACT

**Background:** The objective of this study was to compare and analyze the representative opening and closing movement of Tai Chi elastic band exercise with the reverse fly movement of elastic band resistance training. The aim was to explore the biomechanical differences between the two exercises and provide theoretical support for the application of Tai Chi elastic band exercise in health intervention.

**Methods:** A total of 26 male participants were recruited and randomly divided into two groups in a 1:1 ratio. There were 13 participants in each Tai Chi elastic band exercise group and elastic band resistance training group. Both groups of participants used an elastic band to perform movement in the experiment. Experimental data were collected using the Vicon infrared motion capture system and Delsys surface EMG system. The AnyBody software was utilized to simulate the creation of a musculoskeletal model for both exercises.

**Result:** The study found that the Tai Chi elastic band exercise group exhibited smaller horizontal abduction angle and flexion angle of the shoulder joint, as well as normalized RMS of the anterior deltoid and triceps brachii, compared to the elastic band resistance training group ($P < 0.01$); the Tai Chi elastic band exercise group exhibited greater elbow flexion angle, elbow flexion torque, and muscle strength of the infraspinatus, coracobrachialis, biceps brachii, brachialis and brachioradialis, compared to the elastic band resistance training group ($P < 0.01$); the Tai Chi elastic band exercise group exhibited smaller horizontal abduction angular velocity of the shoulder joint and a lower normalized RMS of the posterior deltoid, compared to the elastic band resistance training group ($P < 0.05$).

**Conclusion:** (1) The opening and closing movement of Tai Chi elastic band exercise is characterized by a large elbow flexion angle, a small shoulder joint horizontal angle and flexion angle, and a slow and uniform speed of movement. The reverse fly movement of elastic band resistance training is characterized by a large horizontal abduction angle of the shoulder joint, a large flexion angle of the shoulder joint, a small flexion angle of the elbow joint, and a fast and uneven speed. (2) The opening and closing movement exerts a greater torque on the elbow flexion, while the reverse fly movement exerts a greater torque on the shoulder joint horizontal abduction and external rotation. (3) The opening and closing movement provide greater stimulation to the infraspinatus, coracobrachialis, and elbow flexor, while the reverse fly movement provides greater stimulation to the posterior deltoid, anterior deltoid, subscapularis, and elbow extensor. In summary, the variation in joint angle, joint angular velocity, and hand position could be the factor contributing to the differences in joint torque and muscle activity between the opening and closing movement of Tai Chi elastic band exercise and the reverse fly movement of elastic band resistance training.

# INTRODUCTION

Recently, the topic of 'health' has gained significant attention worldwide. Exercise, as a key component of a healthy lifestyle, plays a crucial role in promoting physical well-being. The latest guidelines, released by the World Health Organization (WHO) on November 25, 2020, emphasize the importance of physical activity, including muscle strengthening, for people of all ages. It is recommended that individuals engage in a certain intensity of aerobic exercise and muscle strengthening exercises on a weekly basis, while also limiting sedentary behavior, in order to enhance overall health (*Bull et al., 2020*). The training effect in resistance training is influenced by the speed of movement. Traditional speed resistance training imposes a greater load compared to slow speed resistance training, yet both types of movement speed resistance training exhibit certain effects on novice individuals (*Lyons & Bagley, 2020*). However, research has shown that high-intensity exercise can lead to joint pain, making it unsuitable for older adults and individuals with poor physical fitness. Therefore, it is necessary to develop resistance training methods that can effectively increase muscle strength without imposing excessive strain on the body (*Yasuda, 2022*).

Resistance bands are a versatile training tool that allows for flexible adjustments in exercise intensity (*Andersen et al., 2010*; *Soria-Gila et al., 2015*), offering the benefits of affordability and portability (*Colado & Triplett, 2008*). Research has demonstrated that resistance band training can effectively enhance muscle strength (*Martins et al., 2013*), joint range of motion (*Funda & Karakoyun, 2022*), peak torque (*Shoepe et al., 2011*), and

normalized electromyographic (EMG) activity (*Andersen et al., 2010*). Consequently, resistance bands present numerous advantages as a training aid. A study comparing the use of resistance bands and dumbbells for the fly and reverse fly movements found that resistance bands provide an unstable resistance training method that can elevate the activation of supporting muscles, this suggests that resistance bands may be a viable alternative to dumbbells for these exercises (*Bergquist et al., 2018*). Tai Chi, a traditional Chinese exercise, is widely recognized for its positive impact on muscle strength (*Jia et al., 2018*; *Su et al., 2015*; *Yang et al., 2021*; *Lu, Hui-Chan & Tsang, 2013*) and cardiopulmonary function (*Nery et al., 2015*; *Fujimoto et al., 2011*; *Hong, Li & Robinson, 2000*). Recent research has explored combining Tai Chi with lower limb resistance band training, demonstrating improved balance in elderly individuals (*Robitaille et al., 2005*). Some researchers have gone a step further by incorporating resistance bands into Tai Chi practice, where participants are asked to execute Tai Chi movement while using resistance bands. The findings suggest that this intervention has a notable positive impact on enhancing muscle strength in both the upper and lower body (*Lin et al., 2015*). Based on this, we developed the Tai Chi elastic band exercise, which is derived from the Tai Chi Cardiac Rehabilitation Program, with the opening and closing movement as the core (*Ma et al., 2020*). Previous studies have demonstrated that Tai Chi Cardiac Rehabilitation Program can effectively enhance balance (*Hu et al., 2021*), reduce anxiety, and alleviate depressive symptoms in patients (*Lyu et al., 2022*). However, there is a lack of comprehensive research on the biomechanical features of Tai Chi elastic band exercise. During the 'Tai Chi cardiac rehabilitation program' intervention, it was observed that Tai Chi elastic band exercise resemble elastic band resistance training, but the specific biomechanical characteristics remain unclear. This study aims to explore the biomechanical differences between the opening and closing movement of the Tai Chi elastic band exercise and the reverse fly movement of elastic band resistance training. One group of participants practiced the opening and closing movement of the Tai Chi elastic band exercise, while another group practiced the reverse fly movement of the elastic band resistance training. The research reveals distinct biomechanical characteristics between the two movements and identifies the underlying reasons for these differences. These findings not only contribute to distinguishing between the Tai Chi elastic band exercise and elastic band resistance training but also offer theoretical support for integrating the Tai Chi elastic band exercise into the Tai Chi Cardiac Rehabilitation Program.

The research hypothesis of this article is that the opening and closing movement of the Tai Chi elastic band exercise and the reverse fly movement of elastic band resistance training have different biomechanical characteristics. Additionally, the joint angle, joint angular velocity, and different positions of the hand may cause differences in joint torque, average muscle strength, and normalized RMS (Root Mean Square, RMS) between the opening and closing movement of the Tai Chi elastic band exercise and the reverse fly movement of elastic band resistance training.

## MATERIALS AND METHODS

### Participants

In this study, a total of 26 male participants were recruited from Beijing Normal University and randomly divided into two groups at a 1:1 ratio. One group was assigned to the Tai Chi elastic band exercise group, while the other group was designated as the elastic band resistance training group, with an equal distribution of 13 participants in each group. The inclusion criteria for the study were as follows: (1) right hand as the habitual hand, (2) male gender, (3) no history of upper limb injury within the past 6 months, (4) no participation in strenuous exercise within 24 h prior to the test. There were no significant differences in age, height, and weight between the two groups ($P > 0.05$). Before the experiment, the subjects were informed about the specific experimental process and were required to sign an informed consent form. This experiment was conducted as part of the National Key Research and Development Program of China, specifically focusing on the "Research on the key Technology of personalized precision guidance program for Human Exercise to promote Health". The experiment received approval from the Exercise Science Experimental Ethics Committee of Beijing Sport University (approval number: 2018018H).

In this study, Gpower 3.1 software was used to test the samples after the event. The results indicated that the experimental efficacy was 0.81 ($\beta = 0.81$ and greater than 0.8). Previous studies have shown that test power greater than 0.8 indicates that the sample size is reliable (*Baig & Jaisharma, 2022*). This suggests that the sample size used in this study is reliable. Furthermore, the focus of this article is primarily on analyzing the upper limb part of the opening and closing movement of the Tai Chi elastic band exercise. Surface EMG data was collected during the experiment. In order to avoid any potential influence of physiological differences between genders, only male subjects were selected for this study. The participants' basic information is listed in Table 1.

### Instrumentation

In this study, kinematic data were captured using eight high-precision infrared motion capture systems (Vicon Vantage V5; Oxford Metrics Limited, United Kingdom) with a frequency of 100 Hz. Surface EMG data were acquired using the Delsys surface electromyography test system (Delsys, American) with a sampling frequency of 2,000 Hz.

### Determination of Tai Chi elastic band exercises and elastic band resistance training test movements

The Tai Chi elastic band exercise was developed by our research group (*Jia et al., 2018*). In this exercise, the opening and closing movement are characterized by their simplicity, ease of learning, and effectiveness in promoting physical fitness. Therefore, these movements have been chosen as the central focus of the Tai Chi elastic band exercise. Based on these reasons, it was decided to select the opening and closing movement for this experiment.

To minimize interference from irrelevant factors, the reverse fly movement of shoulder joint doing horizontal abduction, which has a similar movement route and joint

**Table 1 Basic information of the participants in the Tai Chi elastic band exercise group and the elastic band resistance training group (mean ± SD).**

| Body parameters | Groups | | T value | P value |
|---|---|---|---|---|
| | Tai Chi elastic band exercise group (n = 13) | Elastic band resistance training group (n = 13) | | |
| Age (y) | 21.62 ± 2.06 | 22.46 ± 2.47 | −0.948 | 0.353 |
| Height (cm) | 177.69 ± 6.63 | 174.54 ± 4.45 | 1.425 | 0.167 |
| Weight (kg) | 70.96 ± 15.18 | 67.89 ± 12.92 | 0.557 | 0.583 |
| BMI (kg/m$^2$) | 22.39 ± 4.12 | 22.20 ± 3.59 | 0.128 | 0.899 |

**Note:**
  $P > 0.05$ indicates no notable differences in basic information between the two groups.

movement form as the opening and closing movement, was chosen as the test movement for elastic band resistance training.

In order to ensure that each participants could fully master the movements before the test and make the experimental results reproducible, experts in related fields were invited to train the two groups of subjects three times, each session lasting 30 min. The Tai Chi elastic band exercise group focused on practicing the opening and closing movement, while the elastic band resistance training group focused on practicing the reverse fly movement. Following the training, both groups were evaluated by experts and deemed eligible to proceed to the next phase if they met the assessment criteria. Those who did not pass the evaluation continued to practice until they achieved a qualified assessment.

Following the completion of the movement assessment, each participant underwent the 10-Repetitions Maximum (10RM) test to determine the suitable strength of the elastic band. This involved performing an incremental load test until the subject could only complete 10 repetitions maximum (*Fernandes et al., 2022*). After the completion of a 10RM test, a formal test will be conducted 3 days later.

The requirements for the opening and closing movement are as follows: maintain an upright posture, pull the shoulders back, bend the knees, and loosen the hips. The wrists should be in a neutral position with the elastic band wrapped around the outside. Keep the arms and shoulders at the same height, with the fingertips facing inward. Slowly open the arms to their maximum extent and then slowly return to the original position, as shown in Fig. 1. For the reverse fly movement, maintain an upright posture with the chest raised and the jaw slightly retracted. Retract the shoulders and rotate the wrists internally. Hold the elastic band with both hands and raise the arms along with the shoulders. Keep a slight bend in the elbows and open the arms to their maximum extent. Slowly retract the arms back to the original position, as shown in Fig. 2.

## Simulation system

The full-body musculoskeletal model was created using AnyBody 7.2 (AnyBody Technology, Aalborg, Denmark), a software used for simulating human motion and mechanical behavior. The human body model in the AnyBody 7.2 software has been extensively validated through numerous experiments and has demonstrated high accuracy

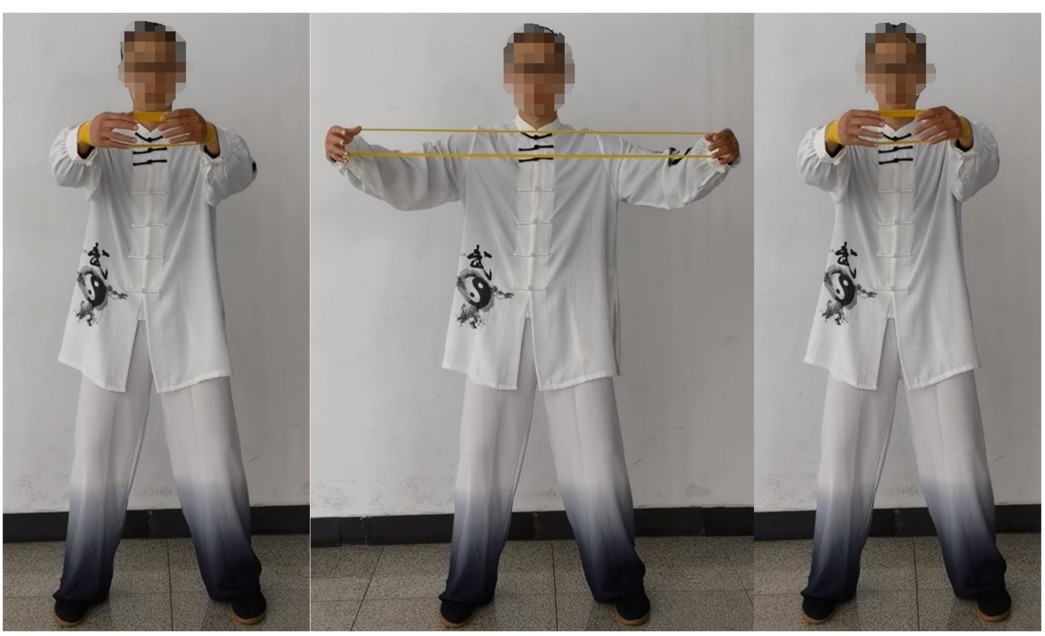

**Figure 1 The process of opening and closing movement of Tai Chi elastic band exercise.**

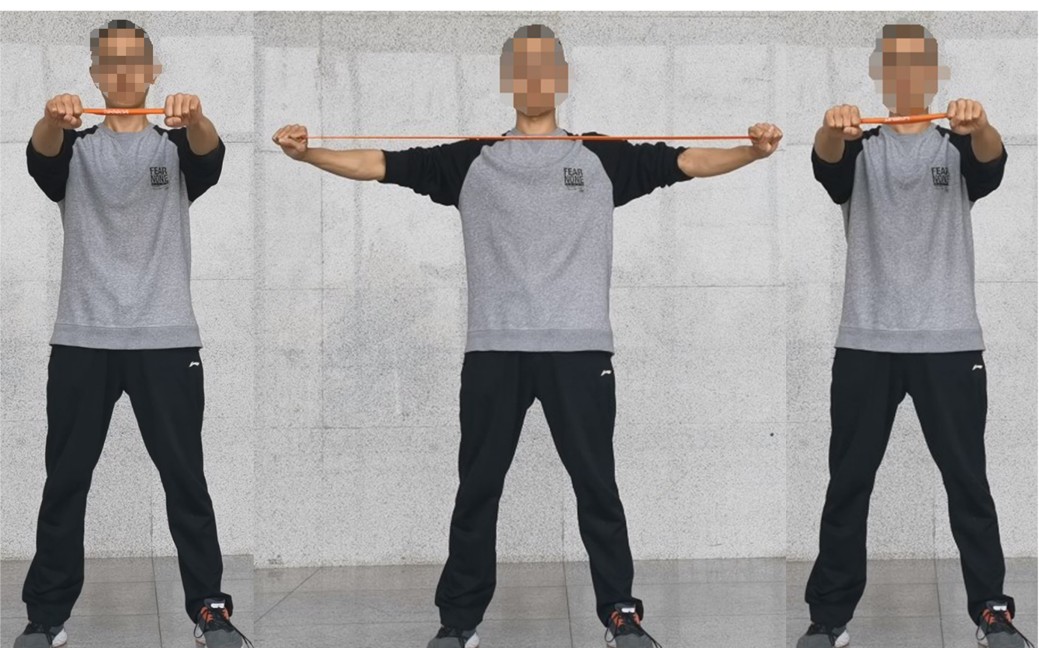

**Figure 2 The process of reverse fly movement of elastic band resistance training.**

and reliability in predicting muscle and joint reaction forces in various scenarios (*Nimbarte et al., 2013*; *Mubarrat & Chowdhury, 2023*; *Engelhardt et al., 2020*).

The musculoskeletal model constructed by the AnyBody software is a standard multi-body dynamics model that includes rigid components (such as human bones or

external objects), kinematic actuators (representing body movements), and force/torque actuators (representing muscles) (*Li et al., 2023a*). The motion-induced forces and moments are simulated using a multi-body dynamics simulation approach.

## Testing protocols

The experiment was conducted in the virtual simulation laboratory of Qiu Jiduan Gymnasium at Beijing Normal University. Firstly, the infrared motion capture device was calibrated to ensure its proper functioning and to ensure that each camera could capture the subject's complete motion. The equipment debugging steps included shielding excess reflective points, calibrating the spatial coordinate system, and calibrating the capture center points. The surface EMG equipment was then opened and tested for normal operation. The required electrode sheet was selected and prepared for testing.

The participants wore uniform shorts and black experimental socks to minimize errors caused by clothing movement. The Vicon infrared motion capture system required attaching 41 motion capture reflection points to the skeletal marker points of the Plug-in-Gait model (*Bassani et al., 2017*) (Fig. 3). The skin surface was cleaned with a 75% alcohol cotton swab, and any hair on the abdomen was removed with a razor to prevent interference with the surface EMG signal. In this experiment, nine surface muscles were selected: biceps brachii, triceps brachii, anterior deltoid, posterior deltoid, trapezius, infraspinatus, pectoralis major, latissimus dorsi, and serratus anterior. The alignment of surface EMG electrodes was consistent with the orientation of muscle fibers. The specific positions are shown in Fig. 3 and Table 2. Prior to the test, each subject underwent a maximum voluntary contraction (MVC) test to normalize any differences in muscle contraction ability between subjects. The location of the EMG electrode and the MVC test method are shown in Table 2.

The procedure of MVC test: After a 10-min warm-up activity, the participants were instructed to perform various muscle test movements as outlined in Table 2. All movements were initiated upon hearing the command "start," with gradual increase in contraction strength aiming to achieve maximum power within approximately 2 s. Subsequently, subjects maintained this maximum contraction for a duration of 3 s before gradually reducing their exertion. Verbal rewards were provided to all subjects during the MVC test (*Andersen et al., 2010*).

## Data collection and analysis

The focus of this article is on analyzing the movements on the right side, specifically examining the movement characteristics of the right upper limb and trunk. Participants began the movements at the center point of the Vicon infrared motion capture system upon hearing the start signal. The Tai Chi elastic band exercise group performed the opening and closing movement, while the elastic band resistance training group performed the reverse fly movement. Each movement was performed twice and data was collected accordingly. The opening and closing movement took 10–12 s to complete, while the reverse fly movement took 3–5 s.

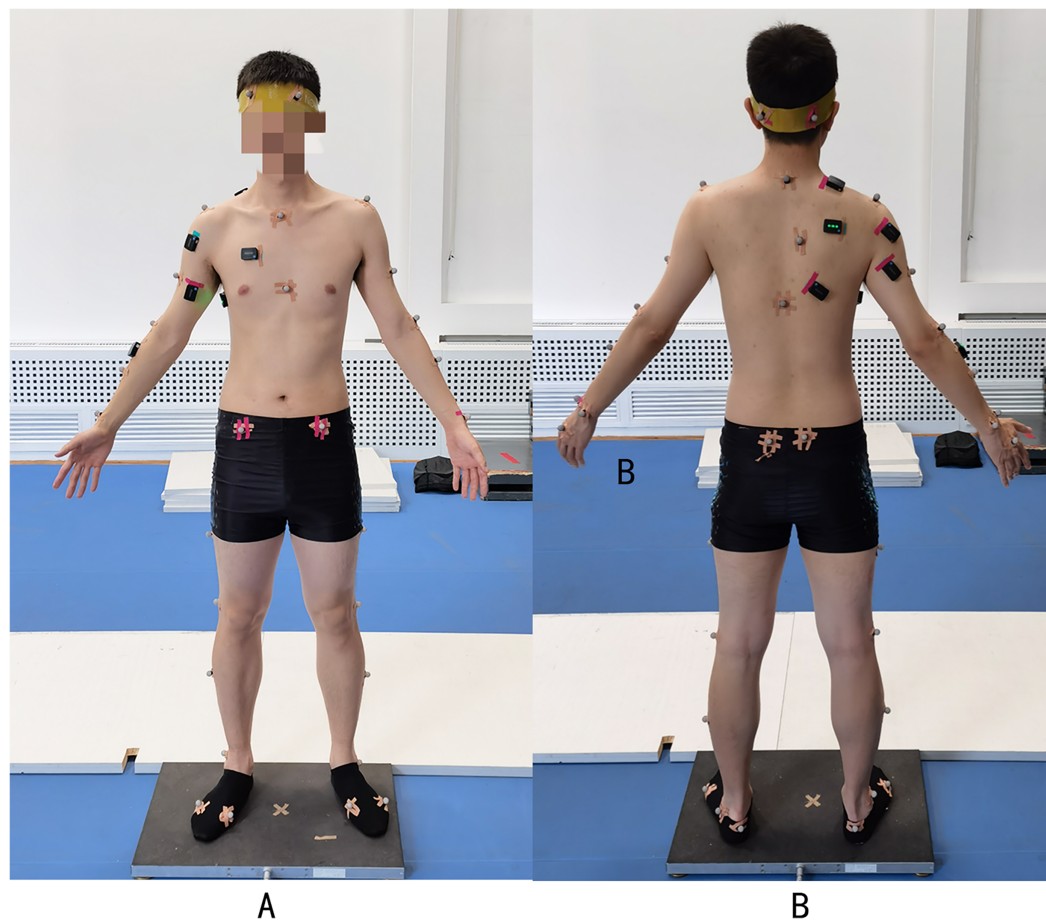

**Figure 3 Landmarks and EMG electrode paste positions.** (A) Front; (B) back.

**Table 2 The positioning of adhesive electrodes for surface EMG and the method for MVC tests.**

| Muscle | Paste position | MVC test method |
|---|---|---|
| Biceps brachii | At the distal 1/3 of the line connecting the acromion point and the ulnar fossa, the electrode direction was parallel to the line connecting the acromion point and the ulnar fossa (*YuZhang, 2015*). | Posture: sitting position<br>Shoulder flexion was 90° and elbow flexion was 90°.<br>The participants applied inward resistance on the medial side of the wrist, and the subjects flexed their forearms (*YuZhang, 2015*). |
| Triceps brachii | The point at 1/2 of the line between the posterior crest of the acromion and the olecranon fossa is two fingers wide inward, and the electrode direction is parallel to the line between the posterior crest of the acromion and the olecranon fossa (*YuZhang, 2015*). | Posture: sitting position<br>Shoulder flexion was 90° and elbow flexion was 90°. Participants applied forward resistance on the lateral side of the wrist, and the subjects' forearms were stretched (*YuZhang, 2015*). |
| Anterior deltoid | The electrode direction is parallel to the direction of the acromion to the thumb at the position of 1 finger width in the front of the acromion (*YuZhang, 2015*). | Posture: sitting position<br>The shoulder joint abduction was 90°. The participant applied downward resistance above the humerus, and the subject's straight arm abducted for isometric contrmovement (*YuZhang, 2015*). |

| Table 2 (continued) | | |
|---|---|---|
| Muscle | Paste position | MVC test method |
| Posterior deltoid | The position of the width of the two fingers after the acromion angle, the direction of the electrode from the acromion to the thumb (*YuZhang, 2015*). | Posture: sitting position<br>Shoulder abduction was 90 ° and elbow flexion was 90 °. Shoulder slightly internal rotation. The resistance applied by the participant to the posterolateral surface of the arm in the direction of adduction and flexion was equidistantly contracted by the participant (*Patterson, Dickerson & Ribeiro, 2020*). |
| Infraspinatus | The electrode was placed 4 cm below the spine of the scapula, and the outside was placed above the subscapular fossa (*YuZhang, 2015*). | Posture: sitting position<br>Shoulder abduction 90°, elbow external rotation 90°. Participants perform equidistant contrmovement against the resistance in the internal rotation direction (*Patterson, Dickerson & Ribeiro, 2020*). |
| Latissimus dorsi | The electrode was placed 4 cm below the tip of the subscapular tip, which is half the distance between the spine and the lateral edge of the trunk. They are oriented at a slight oblique angle of about 25° (*Patterson, Dickerson & Ribeiro, 2020*). | Posture: prone<br>Shoulder adduction and extension and internal rotation. The subjects were subjected to isometric contrmovement of the resistance in the abduction and flexion directions of the forearm (*Patterson, Dickerson & Ribeiro, 2020*). |
| Serratus anterior | The electrode was placed below the armpit, at the tip of the scapula and in front of the latissimus dorsi (*Patterson, Dickerson & Ribeiro, 2020*). | Posture: sitting position<br>Shoulder flexion to 120–130°, and slight abduction. Under the movement of resistance on the back surface of the arm, the participants contracted equidistantly along the extension direction. There is also slight pressure on the lateral edge of the scapula (*Patterson, Dickerson & Ribeiro, 2020*). |
| Pectoralis major | Medial margin of axillary fold, 2 cm below clavicle (*Li & Wang, 2015*). | Posture: prone<br>The body was in a straight line, and the elbow flexion was 90°. The participant applied downward resistance on the back, and the participant performed an upward isometric contrmovement (*YuZhang, 2015*). |
| Trapezius | Placed in the middle point of the electrode along the acromion to the seventh cervical vertebra (*YuZhang, 2015*). | Posture: sitting position<br>The arms abducted 90°, the participant applied downward resistance on the arm, and the experimenter performed an upward shoulder lift (*YuZhang, 2015*). |

## Kinematic and kinetic data

The Vicon Nexus software from the Vicon infrared motion capture system is used for data collection. Motion data of reflective markers on the subjects is then modeled using the Plug-in-Gait model. The resulting kinematic data C3D file was imported into AnyBody7.2 simulation software for calculation. Two skeletal muscle simulation models for horizontal abduction of the whole body shoulder joint were established (Fig. 4). The calculation process involved reflection marker optimization, kinetic calculation, and inverse kinetic calculation. After the calculation, data on joint angle, joint torque, joint velocity, and muscle strength in the exercise state were obtained. The joint angle, joint velocity and joint torque corresponding to the maximum peak state of the exercise were imported into Excel for further analysis. The average muscle strength during the exercise was divided by body

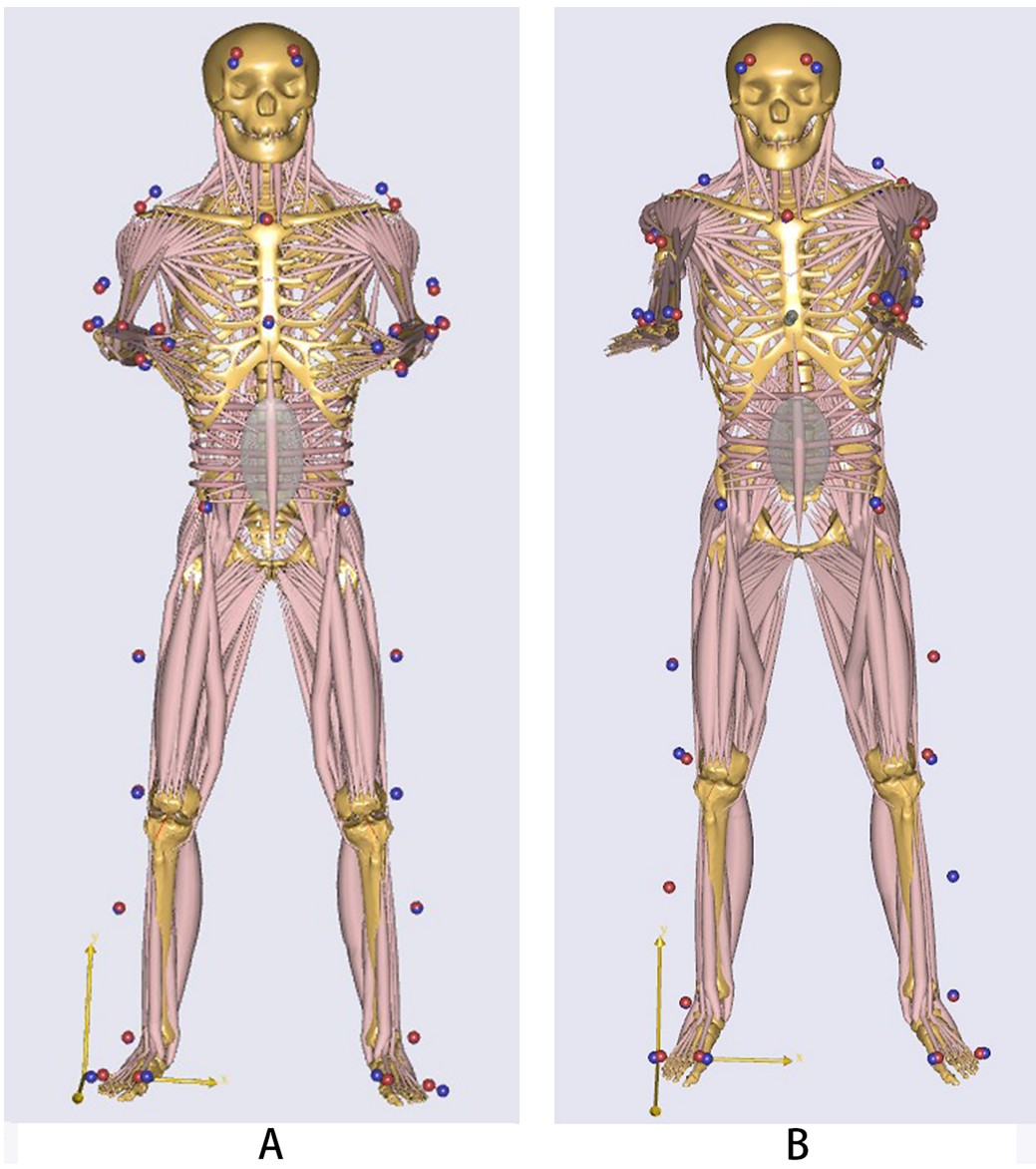

**Figure 4 Skeletal muscle simulation diagram of two elastic band exercises.** (A) Musculoskeletal simulation model for the opening and closing movement of Tai Chi elastic band exercise; (B) musculoskeletal simulation model for the reverse fly movement of elastic band resistance training.

weight and multiplied by 100% (*Li et al., 2023b*). Standardized data can be used for comparison between different groups.

## Surface EMG data

Surface EMG data was collected using EMGworks Acquisition software. The EMGworks Analysis software of the Delsys surface EMG acquisition system was utilized to process the surface EMG signal data, The original EMG signal data during the entire exercise process was filtered and corrected. The filtering range of 20–400 Hz to eliminate low-frequency clutter and high-frequency spikes, resulting in a smoother EMG signal (*Li et al., 2023b*).

Subsequently, time domain analysis was conducted on the processed EMG data to obtain the RMS of each muscle. By measuring the MVC value of each muscle in the subjects, the EMG signal was normalized to account for differences in muscle ability among individuals, facilitating the comparison of EMG data across different groups of subjects (*Reaz, Hussain & Mohd-Yasin, 2006*).

## Statistical analysis

In this study, SPSS 20.0 software was employed to process the experimental data. The data are expressed as the mean ± standard deviation (mean ± SD). The normality of data was tested to determine if the data followed a normal distribution. For data conforming to a normal distribution, the independent sample T test was utilized to analyze group differences. Conversely, for data that did not conform to a normal distribution, the Mann-Whitney U test was used. $P < 0.05$ represents a significant difference, $P < 0.01$ represents a very significant difference.

# RESULTS

## Comparison of joint angles between the opening and closing movement of Tai Chi elastic band exerciset and the reverse fly movement of elastic band resistance training

As show in Table 3, the maximum horizontal abduction angle of the shoulder joint and the maximum flexion angle of the shoulder joint in the Tai Chi elastic band exercise group was smaller compared to the elastic band resistance training group ($P < 0.01$), and the maximum flexion angle of the elbow joint in the Tai Chi elastic band exercise group was greater than that in the elastic band resistance training group ($P < 0.01$).

## Comparison of joint angular velocity between the opening and closing movement of Tai Chi elastic band exercise and the reverse fly movement of elastic band resistance training

As show in Table 4, the maximum horizontal abduction angular velocity of the shoulder joint is smaller in the Tai Chi elastic band exercise group compared to the elastic band resistance training group ($P < 0.05$). Moreover, the standard deviation of the Tai Chi elastic band exercise group is lower, indicating more consistent results.

## Comparison of joint torque between the opening and closing movement of Tai Chi elastic band exercise and the reverse fly movement of elastic band resistance training

As shown in Table 5, the shoulder abduction torque and shoulder external rotation torque are smaller in the Tai Chi elastic band exercise group compared to the elastic band resistance training group ($P < 0.01$), and the elbow flexion torque is larger in the Tai Chi elastic band exercise group compared to the elastic band resistance training group ($P < 0.01$), Additionally, the internal rotation torque of the elbow joint is $-0.00026 \pm 0.00094$ in the Tai Chi elastic band exercise group, while it is $0.00038 \pm 0.00099$ in the elastic band resistance training group.

**Table 3 Mean ± standard deviation (mean ± SD) of the joint angle in the Tai Chi elastic band exercise group and the elastic band resistance training group.**

| Joint angle unit. (°) | Group | | T value | P value |
|---|---|---|---|---|
| | Tai Chi elastic band exercise group (n = 13) | Elastic band resistance training group (n = 13) | | |
| Horizontal abduction angle of shoulder joint | 62.38 ± 13.02 | 94.75 ± 12.53 | −6.459 | 0.000** |
| Shoulder flexion angle | 70.35 ± 11.39 | 93.65 ± 13.64 | −4.730 | 0.000** |
| Elbow flexion angle | 55.53 ± 15.53 | 14.08 ± 9.14 | 8.293 | 0.000** |

Note:
** Represents the comparison between the Tai Chi elastic band exercise group and the elastic band resistance training group, $P < 0.01$.

**Table 4 Mean ± standard deviation (mean ± SD) of joint angular velocity in the Tai Chi elastic band exercise group and the elastic band resistance training group.**

| Angular velocity unit. (°/s) | Group | | Z value | P value |
|---|---|---|---|---|
| | Tai Chi elastic band exercise group (n = 13) | Elastic band resistance training group (n = 13) | | |
| Horizontal abduction angular velocity of shoulder joint | 22.68 ± 11.36 | 65.37 ± 78.99 | −2.282 | 0.022* |

Note:
* Represents the comparison between the Tai Chi elastic band exercise group and the elastic band resistance training group, $P < 0.05$.

**Table 5 Mean ± standard deviation (mean ± SD) of joint torque in the Tai Chi elastic band exercise group and the elastic band resistance training group.**

| Unit of joint torque (Nm/BW) | Group | | T/Z value | P value |
|---|---|---|---|---|
| | Tai Chi elastic band exercise group (n = 13) | Elastic band resistance training group (n = 13) | | |
| Shoulder abduction torque | 0.08 ± 0.01 | 0.11 ± 0.02 | −5.071 | 0.000** |
| Shoulder flexion torque | 0.08 ± 0.01 | 0.08 ± 0.01 | −1.226 | 0.232 |
| Shoulder external rotation torque | 0.02 ± 0.01 | 0.04 ± 0.02 | −3.355 | 0.005** |
| Elbow flexion torque | 0.03 ± 0.00 | 0.02 ± 0.01 | 4.058 | 0.001** |
| Elbow internal rotation torque | 0.00 ± 0.00 | 0.00 ± 0.00 | −1.667 | 0.108 |

Note:
** Represents the comparison between the Tai Chi elastic band exercise group and the elastic band resistance training group, $P < 0.01$.

### Comparison of average muscle strength between the opening and closing movement of Tai Chi elastic band exercise and the reverse fly movement of elastic band resistance training

As shown in Table 6, the average muscle strength of posterior deltoid and subscapularis in the Tai Chi elastic band exercise group was lower than that in the elastic band resistance training group ($P < 0.05$); the average muscle strength of infraspinatus, coracobrachialis, biceps brachii, brachioradialis, and brachialis in the Tai Chi elastic band exercise group was greater than that in the elastic band resistance training group ($P < 0.01$); the anterior

**Table 6 Mean ± standard deviation (mean ± SD) of the average muscle strength in the Tai Chi elastic band exercise group and the elastic band resistance training group.**

| Muscle strength unit(N/BW) | Group | | T/Z value | P value |
|---|---|---|---|---|
| | Tai Chi elastic band exercise group (*n* = 13) | Elastic band resistance training group (*n* = 13) | | |
| Posterior deltoid | 0.39 ± 0.13 | 0.57 ± 0.19 | −2.792 | 0.010* |
| Middle deltoid | 0.93 ± 0.10 | 1.02 ± 0.13 | −1.916 | 0.067 |
| Supraspinatus | 0.19 ± 0.06 | 0.24 ± 0.06 | −1.892 | 0.071 |
| Infraspinatus | 1.16 ± 0.18 | 0.79 ± 0.25 | 4.335 | 0.000** |
| Teres minor | 0.32 ± 0.08 | 0.20 ± 0.29 | 1.510 | 0.144 |
| Anterior deltoid | 1.22 ± 0.10 | 1.35 ± 0.10 | −3.309 | 0.003** |
| Pectoralis major | 0.20 ± 0.09 | 0.20 ± 0.09 | −0.047 | 0.963 |
| Latissimus dorsi | 0.46 ± 0.18 | 0.41 ± 0.48 | 0.400 | 0.693 |
| Teres major | 0.00 ± 0.00 | 0.01 ± 0.03 | −0.128 | 0.898 |
| Subscapularis | 0.00 ± 0.00 | 0.02 ± 0.04 | −2.333 | 0.020* |
| Coracobrachialis | 0.38 ± 0.07 | 0.07 ± 0.07 | 12.058 | 0.000** |
| Biceps brachii | 0.34 ± 0.12 | 0.16 ± 0.06 | 4.945 | 0.000** |
| Brachialis | 0.73 ± 0.20 | 0.30 ± 0.24 | 5.088 | 0.000** |
| Brachioradialis | 0.11 ± 0.03 | 0.04 ± 0.03 | 6.618 | 0.000** |
| Pronator teres | 0.07 ± 0.04 | 0.06 ± 0.02 | 0.325 | 0.749 |
| Triceps brachii | 0.00 ± 0.00 | 0.07 ± 0.09 | −3.513 | 0.000** |
| Anconeus | 0.00 ± 0.00 | 0.00 ± 0.00 | −4.026 | 0.000** |

Notes:
* Represents the comparison between the Tai Chi elastic band exercise group and the elastic band resistance training group, *P* < 0.05.
** Represents the comparison between the Tai Chi elastic band exercise group and the elastic band resistance training group, *P* < 0.01.

deltoid, triceps brachii, and anconeus were smaller in the Tai Chi elastic band exercise group than in the elastic band resistance training group ($P < 0.01$). In the Tai Chi elastic band exercise group, the subscapularis value was 7.99E-12 ± 1.95E-11, the teres major value was 2.46E-11 ± 6.02E-11, the anconeus value was 8.97E-10 ± 3.23E-9, the triceps brachii value was 6.30E-04 ± 1.69E-3. In the elastic band resistance training group, the anconeus value was 1.76E-3 ± 3.76E-3, and the four muscles were 0 after retaining two decimal values.

## Comparison of normalized RMS between the opening and closing movement of Tai Chi elastic band exercise and the reverse fly movement of elastic band resistance training

As shown in Table 7, the normalized RMS of the triceps brachii and anterior deltoid in the Tai Chi elastic band exercise group were smaller than those in the elastic band resistance training group ($P < 0.01$); the normalized RMS values of posterior deltoid in the Tai Chi elastic band exercise group were smaller than those in the elastic band resistance training group ($P < 0.05$); the normalized RMS of biceps brachii in the Tai Chi elastic band exercise group was 0.073, and that in the elastic band resistance training group was 0.065. The

**Table 7 Mean ± standard deviation (mean ± SD) of the normalized RMS in the Tai Chi elastic band exercise group and the elastic band resistance training group.**

| Normalized RMS unit (%). | Group | | T/Z value | P value |
|---|---|---|---|---|
| | Tai Chi elastic band exercise group (n = 13) | Elastic band resistance training group (n = 13) | | |
| Biceps brachii | 0.07 ± 0.13 | 0.07 ± 0.08 | −0.284 | 0.776 |
| Triceps brachii | 0.37 ± 0.17 | 0.80 ± 0.34 | −4.038 | 0.000** |
| Anterior deltoid | 0.17 ± 0.11 | 0.34 ± 0.13 | −3.677 | 0.001** |
| Posterior deltoid | 0.37 ± 0.21 | 0.60 ± 0.24 | −2.599 | 0.016* |
| Trapezius | 0.19 ± 0.16 | 0.33 ± 0.27 | −1.632 | 0.116 |
| Pectoralis major | 0.07 ± 0.11 | 0.12 ± 0.20 | −0.026 | 0.980 |
| Infraspinatus | 0.63 ± 0.31 | 0.62 ± 0.28 | 0.009 | 0.993 |
| Latissimus dorsi | 0.16 ± 0.13 | 0.18 ± 0.21 | −0.692 | 0.489 |
| Serratus anterior | 0.28 ± 0.19 | 0.27 ± 0.16 | 0.203 | 0.841 |

Notes:
* Represents the comparison between the Tai Chi elastic band exercise group and the elastic band resistance training group, $P < 0.05$.
** Represents the comparison between the Tai Chi elastic band exercise group and the elastic band resistance training group, $P < 0.01$.

normalized RMS of biceps brachii in the Tai Chi elastic band exercise group was greater than that in the elastic band resistance training group ($P > 0.05$).

## DISCUSSION

### Analysis of joint angles between the opening and closing movement of Tai Chi elastic band exercise and the reverse fly movement of elastic band resistance training

The angle of joint movement is related to the range of joint movement (*Steinberg et al., 2006*), reflecting the characteristics of the movement. The Tai Chi elastic band exercise was initially developed based on the theory and movements of Tai Chi. It incorporates numerous concepts and movements from Tai Chi. In the Tai Chi elastic band exercise, the arm and body move along an arc line similar to Tai Chi (*Wang, 2020*), maintaining stability and avoiding imbalances or lifting the body. The body shape incorporates essential movements of Tai Chi, such as pulling back the chest, focusing on the virtual spirit, and dropping the shoulders and elbows.

In this study, we compared the flexion angles of the shoulder joint in the opening and closing movement of Tai Chi elastic band exercise and the reverse fly movement of elastic band resistance training. We found that the flexion angle of the shoulder joint is smaller in the opening and closing movement of Tai Chi elastic band exercise compared to the reverse fly movement of elastic band resistance training. This smaller flexion angle helps maintain the chest in an 'arc' state. On the other hand, the elbow flexion angle is greater in the opening and closing movement compared to the reverse fly movement. This increased elbow flexion angle allows the arm to be in a 'circular' state. Additionally, the horizontal abduction angle of the shoulder joint is smaller in the opening and closing movement of the Tai Chi elastic band exercise compared to the reverse fly movement of resistance training. By having a smaller horizontal abduction range, the 'arc' state formed by

completely opening and destroying the arm and chest is avoided. Therefore, the opening and closing movement is characterized by a large elbow flexion angle, small shoulder flexion angle, and small horizontal abduction angle of the shoulder joint, which reflects the movement shape of 'including chest and back pulling' (*Lou, 2023*) and 'sinking shoulder and dropping elbow' (*Li et al., 2003*). On the other hand, in the reverse fly movement of elastic band resistance training, the elbow flexion angle is 14.08°, indicating that the arm is almost straight, the shoulder flexion angle is 93.05°, and the horizontal abduction angle is 94.75°, indicating that the arm and the shoulder joint have formed a vertical relationship. Therefore, the reverse fly movement of elastic band resistance training is characterized by a small elbow flexion angle, large shoulder flexion angle, and large horizontal abduction angle of the shoulder joint, which reflects the movement shape of 'sinking shoulder and standing chest, arm extension'. Previous studies have shown that the range of motion of a joint is related to the strength of the corresponding muscle group (*Pinto et al., 2012*). Variances in joint angles during different movements contribute to differences in muscle strength. The opening and closing movement may enhance the flexor strength of the elbow joint and effectively improve its stability by increasing the flexion angle (*Karbach & Elfar, 2017*). The opening and closing movement of the Tai Chi elastic band exercise is characterized by a large elbow flexion angle, a small horizontal abduction angle of the shoulder joint, and a small flexion angle of the shoulder joint. In contrast, the reverse fly movement in elastic band resistance training is characterized by a large horizontal abduction angle of the shoulder joint, a large flexion angle of the shoulder joint, and a small flexion angle of the elbow joint.

## Analysis of joint angular velocity between the opening and closing movement of Tai Chi elastic band exercise and the reverse fly movement of elastic band resistance training

The Tai Chi elastic band exercise emphasizes the importance of slow and uniform movements, similar to Tai Chi (*Lee et al., 2013*). Throughout the entire movement, it is essential to maintain a consistent and slow horizontal abduction of the shoulder joint, along with controlled muscle force and synchronized breathing. This study compared the shoulder joint angular velocity between the opening and closing movement of Tai Chi elastic band exercise and the reverse fly movement of elastic band resistance training. The results showed that the Tai Chi elastic band movement had a slower speed and more uniform speed changes, as indicated by the lower shoulder joint angular velocity and smaller standard deviation. In contrast, the reverse fly movement in resistance training showed faster and more variable speed changes. The slow joint angular velocity enhances the control effect of nerves on muscles, while the slow and uniform movement reduces the strain on irrelevant muscles, promoting more coordinated and evenly distributed muscle force and minimizing the risk of muscle damage. It is worth noting that different movement speeds do not significantly affect muscle strength growth and muscle hypertrophy (*Lyons & Bagley, 2020*; *Schuenke et al., 2012*), indicating that slow resistance exercise is an effective exercise for middle-aged and elderly individuals to enhance muscle strength (*Tsuzuku et al., 2018*). Therefore, the opening and closing movement of the Tai

Chi elastic band exercise is characterized by slow and uniform motion, while the reverse fly movement in elastic band resistance training is characterized by fast and irregular motion.

## Analysis of joint torque between the opening and closing movement of Tai Chi elastic band exercise and the reverse fly movement of elastic band resistance training

Joint torque is a measure of the force generated by joints and muscles during different motion states (*Li et al., 2023b*), indicating the strength of these joints and muscles. The increase in joint torque during exercise not only reflects the movement state of the surrounding muscles but also provides a powerful driving force for joint movement. In this study, we observed that the elbow flexion torque during the opening and closing movement of Tai Chi elastic band exercise was greater compared to the reverse fly movement of elastic band resistance training. However, the shoulder abduction torque and external rotation torque were smaller in the opening and closing movement of Tai Chi elastic band exercise compared to the reverse fly movement of elastic band resistance training. Unlike the elastic band resistance training movement, which requires the elbow joint to be slightly bent, the Tai Chi elastic band exercise requires the elbow joint to be continuously bent during the movement. It is noted that the flexion angle of the elbow during the opening and closing movement is more pronounced compared to the reverse fly movement. The angle of the joint has a direct impact on the torque produced by the surrounding muscles (*Papaiordanidou et al., 2016*). A larger angle results in a greater torque output. Consequently, the significant elbow flexion angle observed during the opening and closing movement may explain the high elbow flexion torque. The elbow joint plays a crucial role in the upper limb as it enables powerful hand and wrist grasp and fine movements (*Fornalski, Gupta & Lee, 2003*; *Farooq & Ali Khan, 2012*), which are essential in our daily life. Therefore, it is vital to practice and enhance elbow flexion ability. The opening and closing movement of the Tai Chi elastic band exercise has a positive impact on the training of elbow flexion ability. Long-term practice can help cushion the external forces exerted on the joint and protect the joint muscles.

*Eustace et al. (2022)* found that increasing joint torque during exercise can enhance the control force of joints on limbs. The opening and closing movement of the Tai Chi elastic band exercise can improve the control ability of the elbow joint by increasing the flexion torque. Research has demonstrated that the impact force on the elbow joint during flexion can be reduced, resulting in a decrease in the peak value of the impact force and a delay in reaching the peak (*Chou et al., 2001*). By maintaining flexion of the elbow joint, the Tai Chi elastic band exercise helps to effectively absorb external force, delaying the onset of peak force and protecting the joint muscles. The larger horizontal abduction torque of the shoulder joint may be attributed to the joint activity angle and joint angular velocity of the reverse fly movement of elastic band resistance training. The experimental results indicate that the angle and angular velocity of the shoulder joint's horizontal abduction are greater when performing the reverse fly movement of elastic band resistance training. Consequently, the elastic band resistance training group exhibits a larger horizontal abduction angle and faster horizontal abduction velocity of the shoulder joint. Previous

studies have shown that joint torque is influenced by the angle and angular velocity of the joint during exercise. As the angle of joint activity increases and the angular velocity becomes faster, the joint torque also increases (*Anderson, Madigan & Nussbaum, 2007*; *Hori et al., 2020*), which explains the larger shoulder joint abduction torque. The larger external rotation torque of the shoulder joint may be due to the wider range of motion required for the reverse fly movement of elastic band resistance training, subjects may exhibit external rotation of the shoulder to compensate. The opening and closing movement of Tai Chi elastic band exercise is characterized by a large elbow flexion torque, while the reverse fly movement of elastic band resistance training is characterized by a large shoulder joint horizontal abduction and external rotation torque. The reason for the difference between the two movements may be attributed to variations in joint angle and joint angular velocity.

## Analysis of the average muscle strength between the opening and closing movement of Tai Chi elastic band exercise and the reverse fly movement of elastic band resistance training

In this experiment, we analyzed the difference in muscle strength between the opening and closing movement of Tai Chi elastic band exercise and the reverse fly movement of elastic band resistance training using the average value of muscle strength. Our findings indicate that the opening and closing movement of Tai Chi elastic band exercise resulted in greater muscle strength in the infraspinatus, coracobrachialis, biceps brachii, brachioradialis, and brachialis compared to the reverse fly movement of elastic band resistance training. On the other hand, the reverse fly movement of elastic band resistance training showed greater muscle strength in posterior deltoid, anterior deltoid, subscapularis, triceps brachii, and anconeus compared to the opening and closing movement of Tai Chi elastic band exercise. The main force muscles responsible for horizontal abduction of the shoulder joint in both exercises were found to be the infraspinatus, middle deltoid, posterior deltoid, and teres minor, which aligns with the research results of *Ito et al. (2023)*. Overall, the experimental results suggest that the difference in shoulder muscles between the two practice exercises mainly lies in the infraspinatus and posterior deltoid. The greater muscle strength observed in the infraspinatus during the opening and closing movement of Tai Chi elastic band exercise may be attributed to two factors. Firstly, the position of the hands in the two practice methods plays a significant role. Research indicates that hand positioning directly impacts muscle force (*Schoenfeld et al., 2013*). The hands in the opening and closing movement of Tai Chi elastic band exercise are in a neutral position, while the reverse fly movement of elastic band resistance training involves a hand varus position. This difference in hand positions may contribute to the disparity in muscle force. Secondly, the degree of infraspinatus strength decreases as the shoulder joint elevation increases (*Sakaki et al., 2013*). Tai Chi elastic band exercises emphasize the concept of 'sinking shoulder and dropping elbow'. Compared to elastic band resistance training, the height of the shoulder and humeral head is lower in Tai Chi exercises, resulting in greater force exerted on the infraspinatus. The infraspinatus is an important part of the rotator cuff muscle and plays a crucial role in stabilizing the shoulder joint (*Sasaki et al., 2019*; *Ha et al., 2013*). In

rehabilitation exercises, enhancing the strength endurance of the rotator cuff muscle group often involves focusing on the infraspinatus. Therefore, exercising the infraspinatus is of utmost importance. The posterior deltoid is primarily responsible for shoulder joint extension and horizontal abduction (*Campos et al., 2020*). The increased muscle strength observed in the posterior deltoid during the reverse fly movement of elastic band resistance training may be attributed to the rapid speed of the movement. The higher velocity of the exercise could result in greater muscle force generation (*Davies et al., 2017*), leading to enhanced strength in the posterior deltoid. Consequently, the opening and closing movement of Tai Chi elastic band exercise provide greater stimulation to the infraspinatus, while the reverse fly movement of elastic band resistance training has a greater impact on the posterior deltoid. The reason for the difference between the two movements may be attributed to the position of the two hands, the flexion angle of the shoulder joint, and the angular velocity of the joint.

When it comes to the main muscle groups involved in shoulder flexion and horizontal adduction, the key difference between the two exercises lies in the activation of the anterior deltoid and coracobrachialis. The coracobrachialis is mainly responsible for shoulder flexion and horizontal adduction, and can make the scapula lean forward and rotate internally during contraction (*Kobayashi et al., 2021*). The range of motion of a joint is influenced by the strength of the associated muscle group, and varying joint angles can result in different muscle forces being exerted (*Pinto et al., 2012*). During the opening and closing movements of the Tai Chi elastic band exercise, proper body alignment involves pulling the chest back, internally rotating the shoulders, and moving the scapula forward. This specific joint angle may necessitate the contraction of the coracobrachialis. Additionally, Tai Chi emphasizes slow and uniform speed (*Zorzi et al., 2015*). Therefore, in the Tai Chi elastic band exercise, the opening and closing movement should be performed with a slow and uniform speed while horizontally abducting and adducting the shoulders, and the coracobrachialis needs to exert continuous force. The main responsibility of the anterior deltoid is shoulder flexion and horizontal adduction (*Campos et al., 2020*). During the reverse fly movement of elastic band resistance training, the flexion angle of the shoulder joint is larger, resulting in more power exerted by the anterior deltoid. Moreover, among the shoulder joint muscles, the subscapularis in the reverse fly movement of elastic band resistance training is more active compared to the Tai Chi elastic band exercise. The subscapularis plays a crucial role in stabilizing the shoulder joint by maintaining the position of the humeral head and offsetting the tension of the deltoid (*Gamulin et al., 2002*; *Kadaba et al., 1992*). Due to the faster speed of elastic band resistance training, the deltoid exerts more power, requiring the subscapularis to maintain the position of the humeral head and counteract the deltoid's force. Consequently, the subscapularis exhibits greater power. Therefore, in the opening and closing movement of Tai Chi elastic band exercise, the coracobrachialis exerts greater force, while the horizontal adductor muscle of the reverse fly movement of elastic band resistance training exerts greater force on the anterior deltoid and subscapularis. The reason for the difference between the two movements may be attributed to variations in joint angle and joint angular velocity.

In the context of the elbow joint, the biceps brachii, brachioradialis, and brachialis are classified as elbow flexors (*Wilk & Andrews, 2001*), while the triceps brachii and anconeus are classified as elbow extensors (*Zhang & Nuber, 2000*). During the opening and closing movement of the Tai Chi elastic band exercise, the elbow joint remains in a flexed position, with a flexion angle of 55.53 degrees. Throughout this movement, the elbow joint experiences resistance from the elastic band, requiring the flexor to continuously exert force. As a result, the biceps brachii, brachioradialis, and brachialis exhibit higher muscle strength in this scenario. On the other hand, during the reverse fly movement of elastic band resistance training, the elbow joint remains in an extended position, with an angle of 14.08 degrees. The resistance provided by the elastic band demands the extensor force of the elbow joint, resulting in greater force exerted by the triceps brachii and anconeus. Consequently, the elbow flexor force is more significant in the opening and closing movement of Tai Chi elastic band exercises, while the elbow extensor force is larger in the reverse fly movement of elastic band resistance training. The reason for the difference between the two movements may be attributed to variations in joint angle.

## Analysis of normalized RMS between the opening and closing movement of Tai Chi elastic band exercises and the reverse fly movement of elastic band resistance training

The normalized RMS can be used to assess the degree of muscle activation (*Tseng et al., 2007*). Based on the research findings mentioned above, it is observed that the anterior deltoid, posterior deltoid, and triceps brachii exhibit lower activation during the opening and closing movement of the Tai Chi elastic band exercise compared to the reverse fly movement of elastic band resistance training. This is consistent with the muscle activity results simulated by AnyBody software. The reduced activation of the anterior deltoid during the opening and closing movement of the Tai Chi elastic band exercise can be attributed to the slower movement speed and smaller flexion angle of the shoulder joint in this exercise. The reduced activation of the triceps brachii may be attributed to the specific body positioning required in Tai Chi, which involves sinking the shoulders and dropping the elbows. This positioning leads to a greater angle of elbow flexion, increased force from the elbow flexors, decreased force from the extensors, and subsequently, lower activation of the triceps brachii as the extensor of the elbow joint. In a previous study, it was found that the neutral position of the shoulder joint, as opposed to the internal rotation state, increased the muscle activity of the posterior deltoid and infraspinatus (*Schoenfeld et al., 2013*). However, this differs from the results of the current study, which can be attributed to the variations in shoulder joint height, humerus head height, and movement speed during the opening and closing movement and the reverse fly movement. Additionally, the surface EMG results did not show a significant difference in the normalized RMS of the infraspinatus and biceps brachii, which contrasts with the muscle strength results simulated by AnyBody software. The difference in the results of the infraspinatus can be attributed to some of the muscles blocking the collection of all surface EMG signals. Similarly, the difference in the results of the biceps brachii is due to its division into long

head and short head, making it difficult for a surface EMG electrode to collect all the signals.

Strengths and limitations of the study: This study holds great significance for the Tai Chi elastic band exercise. The article conducts a scientific analysis of Tai Chi elastic band exercise and establishes a comprehensive musculoskeletal model for Tai Chi elastic band exercise and elastic band resistance training movement. This provides both theoretical and practical guidance for the promotion and development of Tai Chi elastic band exercise. However, there are certain limitations in this study. It only analyzes and compares one movement in the Tai Chi elastic band exercise, and the number of analyzed movements is limited. Furthermore, this study solely focuses on male students, resulting in a limited subject group with a relatively small number of participants. It is recommended to increase both the number of subjects and diversify the subject group in future experiments.

## CONCLUSIONS

This study draws the following conclusions: (1) The opening and closing movement of Tai Chi elastic band exercise is characterized by a large elbow flexion angle, a small shoulder joint horizontal range and flexion angle, and a slow and uniform speed of movement. The reverse fly movement of elastic band resistance training is characterized by a large horizontal abduction angle of the shoulder joint, a large flexion angle of the shoulder joint, a small flexion angle of the elbow joint, and a fast and uneven speed. (2) In terms of joint torque, the opening and closing movement of Tai Chi elastic band exercise produces a large elbow flexion torque, while the reverse fly movement of elastic band resistance training generates a large horizontal abduction torque and external rotation torque of the shoulder joint. (3) Regarding muscle strength, the opening and closing movement of Tai Chi elastic band exercise provide greater stimulation to the infraspinatus, coracobrachialis, and elbow flexor (biceps brachii, brachioradialis, and brachialis). On the other hand, the reverse fly movement of elastic band resistance training offers greater stimulation to the posterior deltoid, anterior deltoid, subscapularis, and elbow extensor (triceps brachii and anconeus). In summary, the variation in joint angle, joint angular velocity, and hand position could be the factor contributing to the differences in joint torque and muscle activity between the opening and closing movement of Tai Chi elastic band exercises and the reverse fly movement of elastic band resistance training.

## ACKNOWLEDGEMENTS

The authors express their gratitude to the Virtual Simulation Laboratory of Beijing Normal University for its invaluable support in data collection.

### Funding

This work was financially supported by the "Beijing Natural Science Foundation (7244482)" of Jianwei Zhang and and the sub-project "Development of Precise Exercise

Prescription for Chinese Population and Construction of Exercise Prescription Database (2018YFC2000604)" under the Key Technology Research of Personalized Precise Guidance Program for Human Movement to promote Health sponsored by Shaojun Lyu of China's National Key Research and Development Program. The funders had no role in study design, data collection and analysis, decision to publish, or preparation of the manuscript.

## Grant Disclosures

The following grant information was disclosed by the authors:

Beijing Natural Science Foundation: 7244482.

Development of Precise Exercise Prescription for Chinese Population and Construction of Exercise Prescription Database: 2018YFC2000604.

## Competing Interests

The authors declare that they have no competing interests.

## Author Contributions

- Mingyu Liu conceived and designed the experiments, performed the experiments, analyzed the data, prepared figures and/or tables, authored or reviewed drafts of the article, and approved the final draft.
- Cuihan Li conceived and designed the experiments, analyzed the data, prepared figures and/or tables, authored or reviewed drafts of the article, and approved the final draft.
- Xiongfeng Li performed the experiments, analyzed the data, prepared figures and/or tables, authored or reviewed drafts of the article, and approved the final draft.
- Jianwei Zhang conceived and designed the experiments, performed the experiments, analyzed the data, authored or reviewed drafts of the article, and approved the final draft.
- Haojie Li conceived and designed the experiments, performed the experiments, authored or reviewed drafts of the article, and approved the final draft.
- Yameng Li conceived and designed the experiments, performed the experiments, analyzed the data, prepared figures and/or tables, authored or reviewed drafts of the article, and approved the final draft.
- Qiuyang Wei performed the experiments, prepared figures and/or tables, and approved the final draft.
- Zaihao Chen performed the experiments, analyzed the data, prepared figures and/or tables, and approved the final draft.
- Jiahao Fu performed the experiments, prepared figures and/or tables, and approved the final draft.
- Yanying Li conceived and designed the experiments, authored or reviewed drafts of the article, and approved the final draft.
- Meize Cui performed the experiments, analyzed the data, prepared figures and/or tables, and approved the final draft.
- Lujia Li analyzed the data, prepared figures and/or tables, and approved the final draft.

- Peng Zhang analyzed the data, prepared figures and/or tables, and approved the final draft.
- Yuerong Huang analyzed the data, prepared figures and/or tables, and approved the final draft.
- Yuxin Ma analyzed the data, prepared figures and/or tables, and approved the final draft.
- Jianan Xu performed the experiments, prepared figures and/or tables, and approved the final draft.
- Shaojun Lyu conceived and designed the experiments, authored or reviewed drafts of the article, and approved the final draft.
- Yunchao Ma performed the experiments, authored or reviewed drafts of the article, and approved the final draft.

### Human Ethics

The following information was supplied relating to ethical approvals (*i.e.*, approving body and any reference numbers):

The experiment received approval from the Exercise Science Experimental Ethics Committee of Beijing Sport University (approval number: 2018018H).

### Data Availability

The data is available at Figshare: Liu Mingyu. Raw data of the new Tai Chi elastic band exercise and elastic band resistance training. figshare. Dataset. https://doi.org/10.6084/m9.figshare.25182362.

### Supplemental Information

Supplemental information for this article can be found online at http://dx.doi.org/10.7717/peerj.17839#supplemental-information.

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
