# Peer review of "Comparative analysis of biomechanical characteristics between the new Tai Chi elastic band exercise for opening and closing movement and elastic band resistance training for the reverse fly movement"

_PeerJ, doi:10.7717/peerj.17839_

## Round 0.1 · original submission · Major Revisions

Dear authors,

The study entitled “Comparative analysis of biomechanical characteristics between the new Tai Chi elastic band exercise for Opening and Closing movements and elastic band resistance training for the Reverse Fly movements” demonstrated interesting findings using an appropriate methodological approach. However, some important points must be clarified in the manuscript. Your article has potential for publication on PeerJ, but the reviewers have requested substantial changes to be made, mainly in methodology and discussion sessions.

Reviewer 1 ·

Basic reporting

1) English is clear throughout the manuscript.

2) Literature reference is appropriate.

3) Figures and tables are appropriate (see comments in the PDF attached).

4) Results are appropriate.

Experimental design

1) Aims and scope are appropriate.

2) The research question is well defined but the relevance and how the study contributes to filling the gaps are not clear (see comments in the PDF attached).

3) Methods mostly describe with sufficient information to be reproducible the study by another investigator (see comments in the PDF attached).

Validity of the findings

1) Although the study presents any obvious impact, the rationale and benefit to literature is not cleary stated (see comments in the PDF attached).

2) Statistical analysis is appropriate for this basic type of descriptive study.

3) The conclusion is quite long and does not present a take-home message to the reader (see comments in the PDF attached).

Additional comments

The present study aimed to compare and analyze the representative opening and closing movements of Tai Chi elastic band exercise with the reverse movements of elastic band resistance training. Based on the results, the authors concluded that the opening and closing movement of Tai Chi elastic band exercises are more suitable for beginners, elderly individuals, and children with lower muscle strength, while the elastic band resistance training reverse fly movement is more suitable for individuals with a sports background and stronger muscle strength. Although the introduction (partly) prepares the audience for the research focus, this reviewer misses evidence from the literature related to biomechanical characteristics of the opening and closing movements using elastic band exercise and with the reverse fly movement in elastic band resistance training (lines 88 to 90 – see specific comments below). What previous studies (not necessarily related to Tai Chi) involving this technique/method have observed? Is the evidence favorable to a positive impact on muscle strength using elastic band exercise? It should be better described in the introduction to justify the hypothesis of the study (that should be presented as well). The experimental procedures are partly (see specific comments below) clearly described and appropriate for this basic type of descriptive study.

Annotated reviews are not available for download in order to protect the identity of reviewers who chose to remain anonymous.

Reviewer 2 ·

Basic reporting

Abstract
. Line 31: Remove the term significantly along with the manuscript when the author has already presented the p value.
. Line 48: Avoid the repetitive use of the word “muscle”.

Materials & methods
. Line 99 and 100: Rewrite.
. Line 122: 100 Hz

Results
. There is no need to repeat the results from the table.
. Line 258: group (P < 0.01)

Discussion
. Line 304: Give space between the text and the references. Check out this comment along with the manuscript.
. Line 340: Eustace…. (Eustace, 2022)
. Line 358: subjects…

Table 1, Table 2, Table 3, Table 4, Table 5, Table 6, Table 7 titles should be rewritten considering the information added in the Table.

Experimental design

The manuscript is well written and very explicative, but some minor changes should be considered.

Validity of the findings

The manuscript can be relevant to the scientific and technical community. There is clear understanding of methods, results and discussion. The authors implemented rigorous methodological analysis, which implied clarity in results and reading discussion.

---

## Round 0.2 · accepted · Accept

Dear Author,

Congratulations! After your diligent work addressing the reviewers' comments, I am pleased to inform you that your manuscript has been accepted for publication in PeerJ. This version is more concise and formal, enhancing clarity and flow.

Reviewer 1 ·

Basic reporting

No comment

Experimental design

No comment

Validity of the findings

No comment

Additional comments

I would like to thank the authors for the great effort to respond to the comments provided by the two reviewers during the reviewing process, especially to the comments of this reviewer. The authors have improved the manuscript significantly from the initial version based on the recommendations from the reviewers. The revised version of the manuscript is very well written; clear, precise, and easy to understand. Regarding my comments raised during the review process, the authors have satisfactorily responded to all my questions and made the necessary changes suggested to the manuscript. The manuscript has important practical implications and should be of great interest to PeerJ readers. I have no further thoughts on the manuscript.